# How to make sense of information about COVID-19 and the vaccine from the authorities? A qualitative study of migrants' experiences in two Swedish communities

Sofie Bäärnhielm[1,2*], Baidar Al-Ammari[1,2], Soorej Jose Puthoopparambil[3], Mattias Strand[1,2], Önver Cetrez[4]

**1** Department of Clinical Neuroscience, Karolinska Institutet, & Center for Psychiatry Research & Stockholm Health Care Services, Stockholm, Region Stockholm, Sweden, **2** Transcultural Centre, Region Stockholm, Sweden, **3** Global health and Migration Unit Department of Women's and Children's Health, Uppsala University, Uppsala, Sweden, **4** Faculty of Theology, Uppsala University, Uppsala, Sweden

\* sofie.baarnhielm@ki.se, sofie.baarnhielm@regionstockholm.se

## Abstract

The COVID-19 pandemic highlighted the communication challenges faced by public authorities in reaching all groups in society. The overall aim of this study is to gain an in-depth understanding of the experiences of non-native Swedish speaking residents in two multicultural municipalities in metropolitan Stockholm regarding the reception and understanding of COVID-19 and vaccination information from authorities. In addition, the aim is to identify sustainable communication approaches for culturally appropriate linguistic translation of health messages. Between May 2022 and December 2023, interviews were conducted with residents (n = 76) in two areas of Stockholm, Sweden. We used qualitative thematic analysis to develop themes. We developed five themes: Understanding and learning about COVID-19 and the vaccine; Conflicting knowledge bases; Whom to trust; Views on official communication and actions; and Preparing the local community for a future health crisis. For the non-native Swedish speaking residents in this study, understanding the authorities' information about COVID-19 and vaccination was complicated, and receiving information in their own mother tongue was important to facilitate understanding. Deciding what sources and information to trust was a multifaceted and difficult task. For authorities to communicate trustworthy information in a public health crisis, such as the COVID-19 pandemic, the findings point to the importance of language, community engagement, authorities working with local health facilities, community actors, trusted leaders and religious organisations, and the value of verbal and culturally adapted information. We suggest that Antonovsky's concept of Sense of Coherence should be considered for successful health crisis communication aimed at behaviour change and the importance of local community preparedness for future health crises. The results suggest that culturally and linguistically sensitive messages that consider

**Data availability statement:** The data that we can share includes the study protocol, which is preregistered on the Open Science Framework (osf.io/rt47j), as well as the interview guide and code tree, which can be found at https://doi.org/10.5281/zenodo.17389414. For data request you can contact associate professor Anna Clara Hollander, PhD, Karolinska Institutet, Department of Global Health. email: anna-clara.hollander@ki.se. Dr. Hollander has no relationship to the present study.

**Funding:** This work was supported by the Swedish Research Council for funding (Dr 2021-06276 to SB). The funders had no role in the study design, data collection and analysis, decision to publish, or preparation of the manuscript.

**Competing interests:** The authors have declared that no competing interests exist.

people's social realities, not least those most vulnerable in society, can support their sense of purpose in changing their behaviour in a crisis, but also inform policy guidance in health care.

## Trial registration

The study protocol has been preregistered on the Open Science Framework (osf.io/rt47j)

## Introduction

The coronavirus disease 2019 (COVID-19) pandemic highlighted the communication challenges that health authorities face in reaching and gaining the trust of all groups in society. These include linguistic, cultural, and structural challenges in reaching and gaining the trust of migrant communities and groups of non-native speakers. This was particularly important in a country like Sweden, where more than 20% of the population is born abroad, with Syria being the most common country of origin [1] and Arabic the second most common language. A disproportionate risk of COVID-19-related morbidity and mortality was seen among migrants in Sweden, with a particularly high risk among migrants from the Middle East and North Africa [2]. Mussino et al. [3] showed that this pattern, which was seen during the first wave of the pandemic, prevailed for some migrant groups during the second wave [3]. Moreover, COVID-19 mortality was probably highly underestimated and underreported among those with a migrant background [3]. Rostila et al. [4] found a much higher risk of COVID-19-related mortality in most migrant groups in Sweden throughout the pandemic, especially during the early phase [4]. Socio-economic status and living conditions were found to explain these inequalities partially, and better access to vaccination was suggested as the most important preventive strategy to mitigate disparities in health-related outcomes.

COVID-19 vaccination uptake was lower among migrant groups in Sweden [4], despite vaccination being offered free to the public. This is perhaps surprising, in light of the observation that three-quarters of first-generation immigrants in Sweden were positive towards COVID-19 vaccination [5]. Interviews with Somali and Syrian immigrants in Sweden showed that they were generally not opposed to vaccination, but rather that they wanted more information before making a decision, preferably from a trusted peer (e.g., a religious leader or community member) or healthcare professional [6]. For groups facing the greatest barriers to care, Svallfors et al. [5] highlights the importance of establishing trust in healthcare providers and government authorities and the importance of providing adequate information about vaccination to enable informed decision-making. For effective communication during a pandemic, the World Health Organization [7] outlines six key principles: accessibility, actionability, credibility, relevance, promptness, and understandability. In many parts of the world, risk communication campaigns during the COVID-19 pandemic targeting

linguistically and culturally diverse groups failed, especially at the beginning of the pandemic [8]. For an effective COVID-19 response, the importance of cultural sensitivity and community engagement is vital [9].

Unlike many other countries, Sweden did not impose a strict mandatory lockdown or curfew during the pandemic but relied heavily on voluntary social distancing. Swedish authorities issued "recommendations" which were expected to guide behavioural change. Recommendations from authorities in Sweden carry a lot of weight and this was particularly the case during the pandemic. The use of the term can be described as a polite euphemism for what people should do. Naturally, any government policy that relies on voluntary behavioural change places great demands on effective communication that is perceived as trustworthy and meaningful. Data from the World Values Survey place Sweden among the countries with the highest levels of interpersonal trust and trust in authorities [10]; however, the same pattern is not necessarily seen across all groups in a society. Migrants with experiences of war and state persecution are more likely to distrust authorities [11]. To adapt behavioural interventions for ethnic minority communities and build trust, Netto et al. [12] identify five principles: using community resources to publicise the intervention and increase accessibility; identifying and addressing barriers to access and participation; developing communication strategies that are sensitive to language use and information needs; working with cultural or religious values that either promote or inhibit behavioural change; and accommodating varying degrees of cultural identification.

### Intercultural COVID-19 communication in Stockholm

In Sweden, the capital Stockholm and surrounding areas were severely affected early in the pandemic. Some multicultural and economically disadvantaged neighbourhoods with large migrant communities, such as Järva in west Stockholm, and Södertälje south of the city, were particularly hard hit. A contributing factor may have been that relevant COVID-19-related information did not reach all groups in Stockholm, especially at the beginning of the pandemic [13]. In addition, it became clear that some groups found it more difficult to comply with government recommendations on social distancing, working from home, and avoiding public transport due to their occupational demands and household overcrowding [14]. When vaccination subsequently began, coverage was low in the same areas and groups [15].

To address the urgent need for effective intercultural COVID-19 communication, Region Stockholm undertook several measures. Region Stockholm is the regional administrative authority responsible for, among other things, healthcare for 2.45 million inhabitants—approximately 25% of the Swedish population—and comprises 26 separate municipalities including the City of Stockholm. As with much of healthcare services during the pandemic, there was no pre-existing plan or protocol for intercultural crisis communication; therefore, a communication plan had to be created from scratch, considering the needs, resources, and knowledge of different actors. As the pandemic progressed, the communication efforts from Region Stockholm about COVID-19 and vaccination became more extensive and included written information translated into many languages on websites and advertisements in newspapers, films, and digital workshops. Moreover, health informers, lay people educated to provide information face-to-face in local communities and networks in their mother tongue, were employed [16], and cooperation with religious congregations and other local communities was initiated. A multilingual COVID-19 telephone service was launched in Stockholm and was later made available nationally [17]. Learning from this major communication effort can provide important lessons for managing future crises, especially those affecting health. However, there has not been any systematic appraisal of this comprehensive multilingual and culturally sensitive communication effort.

### Aims

The overall aim of this study is to gain an in-depth understanding of the experiences of non-native Swedish speaking residents in two multicultural municipalities in Region Stockholm regarding the reception and understanding of information on COVID-19 and vaccination from the authorities. This is initially carried out in an inductive mode with the analysis then deepening through the concepts of resilience, Sense of Coherence, and syndemics. In addition, we aim to identify sustainable communication approaches for culturally meaningful linguistic translation of health information.

## Materials and methods

### Study procedure

The municipalities of Järva and Södertälje were selected as study sites because they have a high proportion of migrants and an overall low socio-economic status. Moreover, they were heavily affected by high infection rates throughout the pandemic, especially during the first wave. Both municipalities also display low vaccination coverage. Although Järva and Södertälje have these similarities, they have different migration histories and have become a new home for different groups of migrants, which has influenced how the areas are shaped. Järva is an economically deprived district in the western part of Stockholm. The Järva area was built in the late 1960s and early 1970s as part of a national programme to solve the housing shortage, consists mainly of apartment blocks, and has a high proportion of migrants (75.6% of the population), compared to Stockholm as a whole (34.9%), including individuals that are either born abroad or have two parents who were born abroad) [18]. Järva includes the area of Rinkeby, where 92.1% of the population has a migrant background [19]. The population in Järva reflects the various waves of migrants and refugees who have arrived in Sweden over the past half-century and currently has a large Arabic-speaking population from the Middle East and a Somali-speaking community. Even before the COVID-19 pandemic, vaccine hesitancy has been identified as a problem in parts of the Järva area [20].

Södertälje has a long history of immigration currently dominated by migrants from the Middle East, in particular an Assyrian/Syrian community (Syrian refers to the specific term *syrianer,* which is used only in Sweden and differs from Assyrian solely in connection to the historical ancestry). The first Assyrians/Syrians officially arrived in Sweden in 1967, leaving the Middle East due to a history of genocide and persecution [21–23]. Södertälje has a population of approximately 100,000 inhabitants, of which 60% have a migrant background. The numbers of Assyrians/Syrians account for at least one third of the population in Södertälje, which is the centre of Assyrian/Syrian presence in Sweden. Assyrians/Syrians are also known for their collective religious and ethnic traditions and rituals, with an Oriental Christian character [24]. They are multilingual, and many are proficient in Swedish in addition to their ethnic language (different emic terms are used, such as Assyrian, Syriac, and Turoyo; the latter is most commonly used in the population we studied). There is also a gender difference, with men holding higher positions [23].

### Sampling and data collection

A purposeful sampling strategy was used, aiming for variation in gender, age, and language [25]. Individual and group interviews were conducted between May 2022 and December 2023 by BAA, SB, and ÖC. Several different strategies were used to recruit residents for interviews. In both geographic areas, local key stakeholders and community persons were contacted and proved to be crucial for recruiting informants. In Södertälje, contacts with churches, community organisations, and ethnic media channels were important, as they reach a large part of the population. Outreach to the Assyrian/Syrian community was facilitated by ÖC, who is a part of the Assyrian/Syrian diaspora and could more easily gain access to key individuals and networks for data collection. In Järva, where the population is mixed, including large Somali and Arabic speaking populations, various community organisations and religious communities were contacted. We also attended a local health week initiative in Järva, as well as the "Järva Week" 2023 and invited residents to participate in the study. We were given access to a room at the local primary care centre. The Järva Week is a large annual national event where politicians, civil society organisations, and the public meet and discuss social and political issues; in 2023, 68,000 persons participated [26].

Written and oral information about the study was given to the participants, they were informed that they could withdraw participation without motivation. Written consent was obtained by the interviewer and was stored separately from the interviews according to the data regulations of Region Stockholm. Sampling did not take place within the framework of healthcare contacts. Most of the people that we approached for an interview were willing to participate, but

not all. Women were generally more interested to be interviewed than men. Those who did not want to be interviewed usually did not give a reason, except for attendants at a local mosque for women who said that they did not want to be interviewed by a Swede (SB) or a Muslim man (BAA). In general, however, access to the Arabic-speaking community in Järva was facilitated by the fact that our team member BAA is a native Arabic speaker from Yemen. The informants interviewed were very open in expressing their views and experiences. However, there were also hesitations. At one point, the researcher (ÖC) was asked by some informants whether they should share their opinion openly with researchers, or if that could create a risk of stereotyping of their community. A total of 76 informants were interviewed: 61 women and 15 men. Of the interviewees, 10 were in the age range 18–30 years, 30 in the range 40–60 years, and 36 were over 60 years. The shortest individual interview lasted only 6 minutes, the longest 1.5 hours, with a mean interview length of 21 minutes. The shortest interviews took place during the Järva Week, as the informants had little time. Of the interviews, 22 were individual, of which four were in Arabic and two in Turoyo. Group interviews were conducted with 54 informants. Two focus group interviews were conducted in Arabic, of these, one was with a Somali interpreter. Two focus groups were held in Turoyo. Two group interviews were conducted with a Somali interpreter and one in Swedish. The interviews in Arabic and Turoyo were translated into Swedish by the interviewer. The informants had the following mother tongues, Arabic, Somali, Turoyo, Kurdish, Turkish, Filipino, Hungarian, Tigrinya, Amarinja, Persian, Norwegian, and, for one informant, Swedish.

The informants were interviewed about how and when they got information about COVID-19 and the vaccine; if their information channels changed over time; if there were difficulties in coming to terms with COVID-19 and vaccination, or conflicting information; what information had been most useful and what they missed; their experiences of language; sources of information; what can create trust and views on future crisis communication. For a detailed interview guide see S1 Text.

## Data analysis

In analysing the data, we used thematic analysis [27] to identify and report themes and patterns within the data. However, as this was a very large dataset, we began by organising the text into broad content areas or index areas [28]. This was done after familiarising ourselves with the material by listening to the interviews and reading the transcripts. The data was then sorted by index areas for the overarching content of the topics discussed [28]. When analysing the index areas, we used an inductive approach, starting with the identification of units of meaning, analysing codes, categories, sub-themes, and finally themes. We started by analysing the two geographical areas separately. However, when we could not see any relevant differences in the information from the two areas, we decided to analyse both areas together. The first levels of coding were done by BAA and SB. The coding and preliminary themes were then discussed with ÖC. Next, all authors were involved, and the interpretation was refined and revised. We wanted to arrive at themes based on the participants' views rather than on pre-established theoretical concepts [29]. The analysis involved refining the index areas in an abductive mode. The software program Nvivo12 was used to support the analysis process [30].

## Ethics and preregistration

This study was conducted in accordance with the ethical standards of the Helsinki Declaration of 1975, as revised in 2008. The communities involved are in a vulnerable situation. Given their relatively small size, careful considerations in fieldwork and data presentation are needed. Furthermore, at the time of the pandemic, there was an infected and stigmatized political, social, and media discourse, which required a careful and responsible approach to fieldwork from our side. For these reasons we have not provided any personal characteristics when presenting the results, as this would highly increase the risk of revealing the identity of the participants. The study was approved by the Swedish Ethical Review Authority (No. 2022-01637-01). The study protocol has been preregistered on the Open Science Framework (osf.io/rt47j).

## Results

The analysis identified five overarching themes. The first four themes are related to making sense of the pandemic situation and health communication about COVID-19. These are: Understanding and learning about COVID-19 and vaccination; Conflicting knowledge bases; Whom to trust; and Views on official communication and action. The fifth theme is of a somewhat different nature, as it relates to the future: Preparing the local community for a future health crisis. In the interviews, several informants responded by talking about their own personal views and experiences. However, many expressed themselves in the context of a collective group affiliation and spoke in terms of "we" or "our people". The group affiliations could refer to ethnic, religious, cultural, and linguistic groups or to being residents of a specific geographical area. The quotes below illustrate this. Informants' narratives of how they tried to understand and make sense of the pandemic were embedded in descriptions of how their lives and the local community were affected and how they coped with the new situation. In some quotes, minor language changes have been made to improve understanding, as some informants did not use their mother tongue.

### Understanding and learning about COVID-19 and vaccination

The theme of understanding and learning about COVID-19 includes informants' understanding of COVID-19, difficulties in understanding, learning about COVID-19, denial of COVID-19, worries and fears related to trying to make sense of the pandemic situation, and the role of language and culture in these efforts. Understanding and learning about COVID-19 and vaccination was an active and interactive process for the informants, except for those who denied the seriousness of the COVID-19 virus. Several informants said that they and their community initially had a relaxed attitude towards COVID-19. Many first regarded it as just a common cold. For many, there was a particular turning point when they suddenly realised that the COVID-19 virus was a serious matter that was causing deaths in their community. In a focus group interview with older Assyrian/Syrian women in Södertälje, one informant described how she learned about COVID-19 by describing a funeral where many attendants were infected and several died:

> In February, we learned about corona, and we thought it was a cold. But in March there was a funeral in Hallonbergen [a parish in the north of Stockholm]. We didn't go, but some friends from the church [in Södertälje] went there with a group. A person who works in the church got sick. Then we were careful and didn't go out. We then became aware. It was reported on TV, the children stopped visiting us and we were afraid. It happened in a short time. They called from the hospital that he was seriously ill [referring to a person who attended the funeral]. (Turoyo speaker, Södertälje, female, in her 60s)

Many informants in both Järva and Södertälje referred to this funeral, which was an early superspreading event. When many relatives and people close to the informants suddenly became seriously ill and died, most informants realised that this was no ordinary cold. An informant in Järva described the situation as suffocating: *"But in the beginning, that was the big bang"* (Somali speaker, Järva, male, in his 50s).

It was not easy for the informants to make sense of the COVID-19 information issued by the authorities. The informants in the focus group of older women were asked if they had ever heard of such a disease before. One Turoyo speaking woman from Södertälje responded by referring to her childhood experiences in her country of origin: "*There used to be the Spanish flu. Many people died then too. It happens from time to time, and people die. There used to be cholera too, and people died.*" The same group of women were asked if they understood what was said when they heard about COVID-19 in the Swedish media. One Turoyo speaking woman from Södertälje replied: "*Yes, some, but not all.*" She was then asked what she did in order to understand better, and she responded: "*I waited for the news in Assyrian. And then I waited for the children, for them to inform me.*"

At the time of the interviews, most informants considered COVID-19 to be a serious virus that required precautions, but not all. Some informants still thought that the COVID-19 virus did not exist or that it was a common cold: "*It's a common*

*illness, like a cold.... I was so close to my children and grandchildren. I travelled by public transport without a mask. I was fine."* (Somali speaker, Järva, female, in her 60s).

Study informants felt that authorities should communicate in a language they could understand. All informants described common problems with understanding Swedish among residents in the local community. Initially, some informants simply did not bother with COVID-19 information because they did not understand what was being said. Others sought information from their country of origin. The medical terminology used in the official information from Swedish authorities was described as particularly difficult to understand. Some informants did not know the meaning of words such as "virus", or the consequences.

*There are old people who only understand Arabic, Somali, or Turkish. They can't understand what this disease means, what measures are, preventive measures. They can't understand. But they understand what disease [kewo in Turoyo, مرض in Arabic, and sjukdom in Swedish] means. If they read or listen in their own language then it is much easier.* (Somali speaker, Järva, male, in his 40s)

During later phases of the pandemic, official information from Swedish authorities was provided in several different languages, which was much appreciated and facilitated understanding.

Some informants, in both geographical areas, expressed life-threatening existential worries and concerns on individual and collective levels about why some ethnic groups were more severely affected than others. In a focus group one informant expressed ambivalent thoughts, ranging from wonder and concern about why her own group was affected, to disbelief that this was the case.

*I don't think there was a threat to a specific group of people. But many, many older people died and some younger people. But I don't think it was specifically targeted at one ethnic group. It was true that it had affected many Assyrians/ Syrians and Somalis*. (Turoyo speaker, Södertälje**).**

The same informant later spoke of the existential worries experienced in the local area of Södertälje, worries that touched on the Assyrian/ Syrian collective history of genocide.

*The world population has also increased and there is disinformation that the elderly, for example, the first group in Sweden [i.e., the first group of Assyrian migrants], were to be purged. Many people were puzzled. As soon as an ambulance left Ronna [a part of Södertälje], there was panic on social media.* (Turoyo speaker, Södertälje, female, focus group).

The Swedish government's recommendations, which were not mandatory in the legal sense, but mandatory in the colloquial/social sense, were sometimes seen as confusing and difficult to understand. However, there was a wide variation in what informants said about the Swedish pandemic strategy and the authorities' use of the term "recommendation". For some, the authorities' regulations were unclear and far removed from their own everyday life during the pandemic. For others, it was something that worried them and that they described as a language and approach that was incomprehensible or difficult to understand for many in their community. Others tried to find a connection between the language of the authorities and their own language and way of speaking. One informant said of the complexity of using, understanding, and responding to the term:

*I have respect for the fact that it's a recommendation and that it's not compulsory as it has been in other countries. But that is also the sacrifice that is made by not providing even more educational information in order to get people to understand. It's typical for Sweden, typically Swedish and it is difficult. People don't really understand it. So, [the word]*

*recommendation is difficult. Internalising it, understanding what a recommendation is, understanding what to say, the great thing about being able to have integrity, not forcing people is difficult to understand."* (Turoyo speaker, Södertälje, female, in her 40s).

In sum, the interactive process of understanding the COVID-19 information provided by the authorities was influenced by many varied factors and affected informants differently.

**Conflicting knowledge bases**

The information channels used by the informants varied widely and sometimes provided conflicting information. Swedish television, radio, social media (mainly Facebook and WhatsApp), fellow community members, neighbours, and family members were common sources of knowledge. Some were active in their search for information, others were more relaxed. Some informants used official Swedish information sources. However, due to differences in available information and language difficulties with Swedish, many informants turned to information from their countries of origin. This often became confusing, as they received information that differed from that in Sweden. One informant said:

*I've listened to a lot of Turkish channels and there was a big difference. There were a lot of restrictions. There were not the same measures taken in different countries.* (Turkish speaker, Södertälje, female, focus group).

Some informants received information at their workplace, especially those working in the healthcare sector: *"Those of us who can speak Swedish were given information by the employer. We were glued to the TV."* (Turoyo speaker, Södertälje) This information was passed on to children and other family members. Family members were an important source of information for many. Older people were informed by children and grandchildren.

Informants particularly valued the possibility of verbal, personal and face-to-face communication with authorities, not least with representatives with a linguistic and cultural background similar to their own, and the opportunity to ask questions. Primary care was central to answering questions about COVID-19 and vaccination. The national medical helpline 1177 was frequently used. Some also used the multilingual national telephone line.

*Yes, I called once and asked when the vaccine would arrive. People say different things. But I'm not a doctor. I asked the telephone line what the different vaccines meant. One was called Sputnik, one Moderna, one Pfizer. They spoke Arabic and I wanted to have that information. […] The information was very helpful* (Somali speaker, Järva, male, in his 60s).

Several informants mentioned that they had spoken to Region Stockholm's health informants in their local communities. With the health informants, they had the opportunity to talk face-to-face and to ask questions in their own languages. The Region Stockholm health informers had also been helpful in showing them where to find official information from the authorities and where to get vaccinated. This helpful approach increased trust in the authorities' information about COVID-19 and vaccination. One informant described the contact with the health informers as follows: *"…there are people standing on the street who are always providing information, distributing information and doing a solid job, God willing."* (Somali and Arabic speaker, Järva, female, in her 30s).

Several informants pointed out that age influences how and where information is sought. Younger generations were described as being more active in seeking information.

*I haven't called 1177 (the national helpline) to be honest. But I know my young ones, they're young adults, they used to call and that's where you notice the difference between the older generation and the younger generation... If you say something, for example, when I meet NN and say it's happened so and so. But my guys, if I tell them… that such and*

*such has happened, where did you hear about it? That's the first question, where can I find it? There has to be some real source that I can refer to, otherwise it doesn't work. That's the difference between us younger and older people.* (Somali speaker, Järva, female, focus group).

In sum, the informants were faced with the challenge of orienting themselves in a situation of diverse and sometimes contradictory information.

## Whom to trust

It was not only accessing understandable and relatable information that was a problem for informants. Deciding whom to trust was also a challenge—Information from which source was more reliable? This was particularly challenging for informants with lack of education and/or limited knowledge of Swedish, especially in relation to vaccination. Some informants interviewed worked in the healthcare sector themselves and were familiar with medical issues. Most informants, however, worked in other fields. Some informants were illiterate and had no formal education. Moreover, navigating official information channels and healthcare in general was challenging for many informants. One informant referred to difficulties in understanding and trusting information from the authorities as part of the challenges of being new in a foreign country, as follows:

*When you move here as an adult, for example, as I did, you don't actually get any information about society. It's almost like coming to the Indian Ocean and you have to swim. […] And […] one can say that […] people simply haven't had enough trust for authorities, and that complicates the communication between them.* (Somali speaker, Järva, female, focus group).

Trust in non-official sources, often credited with misinformation, was associated by some with previous negative experiences with authorities in their countries of origin, as one informant said:

*You come from a country where the government decides something, and you don't think it is good. You think that the government here is as bad as in the country you come from. That the government decides and then you don't obey. Because it's a bluff. And that's not the case here. At least as far as I know. And you carry it with you, that if [the information] comes from higher up, then it's a scam.* (Somali speaker, Järva, female, focus group).

There was a wide variation in the extent to which informants trusted information from public authorities. Health professionals, doctors, and local primary care providers were given high credibility and were often considered trustworthy. This was also the case for informants who criticised the healthcare system during the pandemic and pointed out gaps in care providers' knowledge. In particular, local healthcare professionals providing information in informants' first languages were highly trusted.

For many informants, religious beliefs and the religious community were important in terms of trust and how they and those around them dealt with the spread of the COVID-19 virus and vaccination. The views of, and information provided by, religious leaders, such as imams and priests, were important and trusted by many, even though they were also criticised for being slow to react to the dangers associated with the COVID-19 virus. However, there were also informants critical of relying too much on religion; one person referred to a discussion with a friend about COVID-19 information:

*But he ignored them and said "kit aloho b'shmayo" [in Turoyo, which means there is a God in heaven who protects]. He went to the compound and then fell ill again. He was hospitalised again and died a few days later.* (Turoyo speaker, Södertälje, female, focus group).

The informants had very different views about the COVID-19 vaccine. In the focus group with older women in Södertälje, the informants had a lively, open discussion about their different views, with one informant expressing her trust in doctors from a religious point of view, saying: "*God has given us doctors, could all these doctors be wrong? We should trust them. That's my belief.*" (Turoyo speaker, Södertälje, female, focus group). Some informants were positive about vaccination, others were afraid and hesitant, and still others were afraid of the vaccine that they had taken. Religious considerations could lead to different views and contradictory conclusions, as one informant said: "*I did not take any vaccine at all. There is a God in heaven. I have done well.*" (Turoyo speaker, Södertälje, female, focus group).

Informants often trusted information from their countries of origin more than that provided by the Swedish authorities. In contrast, other informants found the information from their home countries to be very problematic and alarming.

*No, it was a disaster in Turkey. We have a family chat and there they started to be desperate. They started washing everything in chlorine, everything. And there were many other things. It was horrible.* (Turkish speaker, Södertälje, female, focus group).

Several informants expressed their own and others' distrust of political authorities, especially when it came to the vaccine. A Somali-speaking woman in Järva recounted her experience of being with her grandchildren in a local park and discussing the COVID-19 vaccine with others.

*And there I have met many people from different countries. I have had discussions with quite a few. For some, it was obvious, and others were terrified of the side effects. Some said it was just something the government has made up to keep an eye on us, so they didn't want to take it*. (Somali speaker, Järva, female, focus group).

For many, the dilemma of whom to trust was exacerbated by the fact that the healthcare system in their home countries was considered to have reacted differently to the COVID-19 pandemic:

Informants reported that misinformation played an important role in their local communities. When asked what was said about misinformation, one informant replied:

*It was that, well, Bill Gates said that they had implanted microchips and that there was surveillance of the people, and that scared many people and perhaps most of those who had the lowest education and perhaps also limited language skills.*

*Interviewer: So, it was something that was talked about in the area?*

*- It was the most common [thing]. "They're just making it up, you shouldn't take the vaccine because there's a risk of surveillance". Sad, sad, sad. - That you would only live for a short time if you got the vaccine."* (Somali speaker, Järva, female, focus group).

Overall, trust was often built in a personal way, with language, culture, and belonging to the local health system facilitating trust.

## Views on official communication and actions

The Swedish authorities' strategy of not imposing a strict lockdown during the pandemic was appreciated by some, while others found it confusing and difficult to understand, as it differed from that of their countries of origin. A common view among the informants was that it had taken far too long for information to be made available in languages other than Swedish.

"*I saw some posters about COVID-19 in Somali but that was only six months or a year almost after COVID-19 had started. It took quite a long time*" (Somali speaker, Järva, female, in her 20s).

This delay in providing information in languages other than Swedish was seen as problematic. When asked about the implications of the belated access to information in Somali, an informant from Järva said:

*It's difficult. We live here. There are many people from Somalia. Perhaps they are new in Sweden, and they know nothing. If you mention TV1 or TV4 [Swedish television channels], they know nothing. If something happens, it's better to explain right away.* (Somali speaker, Järva, woman, in her 40s).

For some informants, the late provision of information in languages other than Swedish was perceived as a problem on a personal level; for others it was something they saw as mostly affecting people around them. The authorities were also seen as late in providing information to their local communities.

*There was a lack of practical information, written materials for example. In the beginning they didn't offer that. They didn't come out to districts like in Rinkeby. All information was directed towards the affluent areas, towards those who already can and know a lot.* (Arabic speaker, Järva, male, in his 20s).

Many informants felt that the early information about COVID-19 was very vague and that the authorities had not clearly communicated how dangerous the virus could be. The danger was viewed as presented in terms of needs for behavioural changes, which meant social distancing. For several informants, the exact meaning of this was initially difficult to understand and later difficult to implement due to their lived realities. Informants would have preferred a clearer language and communication specifically targeted at their local communities.

*Perhaps they should have called on organisations and churches and mosques and everyone to cease their activities until things improved. Many organisations could then have told their fellow countrymen that this is serious, it's no joke, there are many dead, especially in our district. That could actually have been done. But no.*" (Kurdish speaker, Järva, man, in his 60s).

As residents of Järva and Södertälje, several informants reported that they felt they were discriminated against in many ways. This was also due to their lack of proficiency in Swedish. Several informants emphasised a feeling of powerlessness. Previous experiences of discrimination and lack of agency were reinforced for some during the pandemic.

*During COVID we felt even worse. No one respected us, no one listened to us. If one of us got sick and ended up in the hospital, they wouldn't give us food, they wouldn't give us [medicine], they wouldn't let people visit our sick relatives. And we felt even more worried and even more powerless. And we had felt that powerlessness before, but during the pandemic it became even worse.* (Arabic speaker, Järva, female, focus group).

Other informants, who had quickly understood the seriousness of the COVID-19 virus, took immediate action to minimise the impact of the pandemic in their local areas. This conclusion meant for some that they came into conflict with their communities due to differing views on the potential dangers of the COVID-19 virus. This initially meant conflicts with friends, families, communities, organisations, and religious congregations about how to deal with the situation. These conflicts were felt to have been exacerbated by the late provision of targeted information by the authorities. Severe conflicts also arose later, in relation to vaccination: "*My daughter once walked past the church to go and get vaccinated, but was threatened by those who didn't want to.*" (Turoyo speaker, Södertälje, female, focus group).

However, the informants were very appreciative of the fact that information was later made available in languages other than Swedish. They emphasised the importance of having access not only to written information, but also to people to talk to, ask questions to, and come back to if they had further questions. Informants emphasised the importance of verbal communication and the value of being able to talk and discuss in their mother tongue with others such as children, friends, neighbours, health professionals, and the health informers working for Region Stockholm. The great value of personal face-to-face contact was underscored.

*Yes, I got information from those people, they handed out information [pointing to a health informer present at the Järva Week]. They sit in the Tensta centre [a part of Järva], they sit in different areas. […]. Some of them [referring to people in the area] didn't understand [the information in Swedish provided on] TV, maybe Arabic, Somali, Iraqi, English. They met in the centre; people were standing there handing out pamphlets.* (Somali speaker, Järva, female, in her 40s).

The community-level interventions, carried out by Region Stockholm in the form of multilingual health informers working in their local areas, were highlighted as important and credible sources of information. Region Stockholm also facilitated testing and vaccination with outreach buses that came to local areas, churches, and mosques.

Some informants considered that information came only after their own ethnic group was being blamed for the spread of the virus. They felt that they were seen as guilty for the spread of the virus due to their vulnerability working in the service sector and living in overcrowded apartments.

*After Somalis who work in the service industry, who work as bus drivers, taxi drivers, in nursery schools, had been vilified, then the level of dissemination was very high among them. After the population was demonised, efforts were made to disseminate information in Somali. There was no information before that.* (Somali speaker, Järva, female, in her 20s).

Overall, informants felt that information in languages other than Swedish was important but came too late. This added to the complexity of understanding and managing the pandemic situation.

**Preparing the local community for a future health crisis**

For future pandemics and health crises, informants had various suggestions. Several stressed the importance of preparedness based on the local circumstances, needs, and resources. The significance of addressing social determinants of health specific to a particular geographical area was emphasized. One informant, who highlighted the importance of preparing for future crises said:

*But I guess it's a lesson, that we should not forget about these areas that look like this. It's multicultural, there are many languages and people have different ways of looking at what the authorities say and all that. And then it's even more important that [authorities are] there in the beginning and not at the end.* (Somali speaker, Järva, female, focus group).

Not all informants felt confident about the future; some expressed feelings of hopelessness and resignation about the state of their communities and about the possibility of acknowledging their needs and resources. However, most informants expressed that they thought that things would be handled in a better way in the event of a future pandemic. They suggested that pandemic preparedness involves organised cooperation between authorities, civil society, local leaders, and religious communities. In both geographical areas, informants emphasised the potential of working with existing local organisations in which many residents have confidence. Commenting on the value of working with local organisations, one informant said:

*It means that you can easily reach the citizens through that route. And then there is also an organisation that works with this and that works with the imams. And then the imams and the priests can inform the people.* (Somali speaker, Järva, female, focus group).

To prepare for a future public health crisis, building trust in health messages communicated by public authorities was seen as crucial. Moreover, that this work needs to be anticipatory and ongoing, rather than starting just as a crisis occurs. Ensuring that authorities truly represent the communities they work with was seen as an important aspect in establishing trust.

*The main thing I learned is that the authorities should be multicultural. They should have many different ethnicities [among their staff]. An authority that employs many ethnicities can create trust. Not just those ethnicities that are not present in many areas.* (Somali speaker, Järva, male, in his 20s).

The importance of working with respected local organisations and local community leaders to build trust was also raised. Another aspect addressed was the fact that many of the people living in these geographical areas have fled persecution by the authorities.

*Certain groups, Muslims for example, who have fled from other countries, Christians who have fled from other countries and come to Sweden. The imams, for example, and the priests are their leaders, they trust them 100% compared to the authorities.* (Somali speaker, Järva, female, focus group).

It was suggested that educating imams, priests, and other community leaders could be beneficial, so that they might quickly provide information that residents would trust in the event of a future health crisis: "*They should provide information in organisations and churches, and then it will spread.*" (Turoyo speaker, Södertälje, female, focus group).

There were somewhat divergent views on the question of language. One informant insisted on the importance of Swedish as the main language and emphasised the usefulness of local cultural adaptation of communication interventions: "*Swedish must be the main language and the second language a supplement.*" (Somali speaker, Järva, male, in his 20s) However, most informants stressed the importance of receiving information in one's own mother tongue, as well as the benefits of verbal communication and the opportunity to engage in dialogue that allows questions to be posed to the authorities:

*When it comes to Järva, many people speak different languages and it's very important that [the information] spreads by communicating in their own language. I can't say "nobody", but most people don't like to read, not even I. You grow up with verbal information. You get stuck. Rarely, rarely do I look at a [written information] myself. I prefer that people talk.* (Somali speaker, Järva, male, in his 40s).

In addition to translation, culturally competent outreach strategies were seen as important, as well as engaging and learning from local communities in efforts to prevent and tackle a health crisis. A Kurdish-speaking informant in Järva exemplified this by talking about his own shortcomings in presenting health messages, the value of people such as health informants who spoke with people in their language, knew the culture, and also belonged to the healthcare system. Such a health informant would also be better able to communicate thoughts and cultural values and thus convince better.

Local outreach activities that can meet people face-to-face were seen as helpful. Several informants said that authorities should use a variety of complementary approaches to reach as many people as possible. Informants also mentioned their own efforts to spread information, such as calling everyone in an organisation or informing neighbours, as successful.

*We ourselves had gone around and informed people in the organisation and we said: Listen, now this has happened. How do you want to avoid this, how do you think we should avoid it together in this neighbourhood that we live in? And that has actually worked. Nobody died in the neighbourhood, for various reasons, maybe [protected by] God or something, sure. And I think that's a method that you can use very effectively. It doesn't have to cost society that much.* (Somali speaker, Järva, male, in his 50s).

The presence of healthcare professionals who spoke different languages was also important for communicating health messages in a crisis.

*One's own language is important, and like we all said: Organisations, all kinds of contexts, priests, football coaches, everybody. And then it's important that there's a doctor who speaks your language.* (Arabic speaker Södertälje, female, focus group).

In sum, informants emphasized the importance of language, as well as local, socially and culturally rooted preparedness in dealing with a future public health crisis.

## Discussion

The COVID-19 pandemic highlighted the importance of crisis communication that reaches, and is trusted by, all groups in society. To gain an in-depth understanding of the experiences of non-native Swedish speaking residents in the two multicultural areas of Södertälje and Järva in Stockholm regarding the reception and understanding of COVID-19-related information, we interviewed 76 residents. For the informants in this study, understanding information about COVID-19 and vaccination has in many ways been a complicated process. Language played a central role in the basic understanding of COVID-19-related communication from the authorities. The importance of language in health communication has been addressed in several other studies of the pandemic [31–35]. It is not only language that plays an important role in our informants' understanding of information from the authorities. Socio-economic factors, such as educational level, culture and acculturation, and collective ethnic history were also influential in shaping understanding and creating or erasing trust. The study was carried out in two areas with a high proportion of migrants in the population, many of them with a recent refugee background. These neighbourhoods have many residents with low socio-economic status and low educational level, overcrowded housing, and people tend to be employed in service occupations posing a high risk of infection. The informants in the study received information from various and sometimes contradictory sources, including official information from the Swedish authorities, information from news in their countries of origin, social media, rumours, misinformation, and information passed on by relatives and others. Though this is not a quantitative study, some generational characteristics can be observed. Older informants in particular expressed difficulties in understanding and navigating information from Swedish authorities. They were often reliant on others, such as children, grandchildren, and neighbours. Young people were seen as more active in seeking information. Our findings echo those of Esaiasson et al. [33] on how residents in underprivileged areas in Sweden informed themselves about COVID-19 in the spring of 2020 [33]. These authors found that language was important, but also ethnicity and ethnic identification, and that those who did not experience a sense of belonging in Swedish society primarily relied on other sources of information than Swedish news media.

Swedish authorities chose a slightly different strategy for dealing with the COVID-19 pandemic compared to other countries [34], as Sweden never introduced a complete lock down, relying instead on mutual trust between authorities and citizens. Underprivileged and racialized communities were among the hardest hit by the pandemic in most countries [35]. Understanding how different pandemic strategies and approaches affected vulnerable groups in society is of interest for an inclusive public system that is better prepared for the next crisis. Our findings suggest that avoiding vague and culturally-loaded terms such as "recommendations," would have facilitated understanding among the non-native speakers

in our study. The discourse of giving "recommendations" required reading between the lines. A Norwegian study of COVID-19 communication shows how the use of culturally specific terms can hinder effective health communication and increase inequalities [31].

Another challenge during the COVID-19 pandemic was that due to the rapidly shifting nature of a viral pandemic, COVID-19-related information provided by the authorities sometimes changed quickly. For informants, the changing messages and the variety of information channels used, often with conflicting information, contributed to confusion about whom to trust. For some, trust was built by trying to find relevant information from official channels and the media. This could be official Swedish channels, the country of origin or other sources. However, for many, what was considered a trustworthy source of information was shaped by trust in the person providing the information. Local religious leaders and healthcare professionals, especially those who speak the language of the migrant community, were considered trustworthy by many. Other people connected with the healthcare system who provided information in the informant's mother tongue, such as the health informers, were also seen as trustworthy. Informants emphasised the importance of being able to ask questions, especially older people who preferred talking face-to-face. A study of the devolvement of trust among migrants in Norway during the COVID-19 pandemic underscores that while a pandemic creates more vulnerability, it also creates valuable opportunities for building trust [36]. The authors suggest that trust is fostered through relationships in the host country and that healthcare providers play an important role in building trust, since they typically interact continuously with the community over time.

Notably, this study was conducted in communities in which high levels of vaccine hesitancy have previously been identified [20]. Our findings regarding the challenges for informants in deciding whose information to trust and the value of community-based communication efforts are consistent with existing research on vaccine hesitancy. In 2019, WHO declared vaccine hesitancy as one of the top 10 global health threats [37]. In migrant communities, a review [38] suggests that social exclusion is a main pathway to developing vaccine hesitancy, and that experiences of marginalisation or discrimination may lead migrant communities to distrust health systems in general and vaccination in particular. The importance of local community engagement that is culturally sensitive and that addresses language barriers has been shown to contribute to overcoming vaccine hesitancy [39–41].

As described above, some study informants expressed a general lack of trust in the governing political authorities. Some even believed that the authorities wanted to use the virus or the vaccine to control them or get rid of them. Given that many residents of Järva and Södertälje have fled state persecution or come from war zones with unreliable authorities, such as the Assyrian/Syrian diaspora with historical traumas of persecution and genocide, historical experiences can influence how a current situation of danger and threat is interpreted. One example is how terms such as virus and bacteria (i.e., microbes) have been used to dehumanise minorities and legitimise genocide, as in the case of the Ottoman Empire, and the negative impact this has had on the collective memory of Assyrians/Syrians [42,43]. Collective historical memories can influence how people interpret new health crises, how communication is received, who is trusted and how the situation is dealt with. How collective historical experiences can influence current well-being and coping is captured by the concept of historical trauma. Historical trauma describes the long-term effects of colonisation, cultural repression, and historical oppression and helps to explain persisting inequalities in health and well-being [44]. The consequences of trauma exposure are known to be passed on to offspring across generations [45] and transmission shaped by several factors [46].

Historical trauma and collective traditional coping strategies for dealing with distress may have influenced how COVID-19 crisis communication was perceived. For the process of understanding and accepting new health messages in a crisis situation, Antonovsky's Sense of Coherence (SOC) concept [47] may be useful for individuals as well as social and cultural groups. The SOC concept has three core components: comprehensibility, manageability, and meaningfulness [47]. New health messages in a crisis that are intended to promote behavioural change need to be comprehensible, manageable, and meaningful for recipients. The SOC concept may be particularly relevant in a pandemic, which is by its nature a stressful, socially and existentially disruptive experience. In terms of comprehensibility, this was reflected in our results on

the term *"recommendation"* that requires a culturally specific level of understanding and reading between the lines. The SOC concept of manageability is reflected in the informant's experience of both a lack of practical information and appreciation of practical information when it was given on issues such as testing and vaccination. Problems with manageability are also reflected in their difficulties in implementing health messages due to their lived social realities. For individuals to find authorities' COVID-19 information as meaningful, they need to be able to link the individual event to their basic approach to life [48]. Especially in the beginning of the pandemic, as it was hard for many informants to understand what was said and how to interpret it, the credibility of authorities' information and their advice was not meaningful. The credibility of authority information and the fact that advice is socially manageable make authority information meaningful.

The SOC concept is also linked to intercultural health communication. The Lancet Commission on Culture and Health points out that this is not just about language skills, but also about people's beliefs about what constitutes effective healthcare and their personal ability to prevent illness and influence disease outcomes [49]. They suggest that successful communication in multicultural settings requires community understanding, trust, and continuity. While they appreciate the value of health information technology in care, they underscore that this is not a substitute for face-to-face contact. Increasingly, information on health issues is disseminated through digital platforms and social media. These platforms played an important role during the pandemic, enabling the rapid dissemination of information to large groups of people. At the same time, this has also meant that misinformation and false information could spread quickly and have a major impact globally [50]. For the informants in our study, the extent to which they participated in digital information varied, but it was common to use traditional media from the country of origin, social media, and other digital communication tools. While the approach of turning to information from the country of origin probably had a coping function in dealing with the emergency by gathering information and gaining understanding, it also increased confusion.

For managing future health crises, several informants stressed the importance of local preparedness. Informants had concrete suggestions that underlined the importance of preparedness based on the local situation, needs and resources, and the importance of addressing the specificity of the areas. This emphasis on local crisis communication is in line with findings from a Swedish study of the impact of media coverage on individual behaviours during the COVID-19 pandemic, which found that the impact was greatest when COVID-19 messages were more locally relevant, visible, and factual [51]. Our informants' experience of discrimination and being singled out as responsible for the spread of the COVID-19 virus shows how important it is to counteract the spread of stigmatising narratives during a pandemic. Our findings suggest that during crisis, such as a pandemic, an inclusive and participatory approach can facilitate effective communication. It is important to have robust structures for collaboration and analysis in place before a health crisis occurs.

Vulnerability to COVID-19 cannot be fully explained by individual factors. Airhihenbuwa et al. [9] emphasise the importance of communication that responds to the community as a collective, community engagement, and addresses community risks at least as much as individual risks. They see culture as central to effective risk communication. When approaching risk, resilience has been understood as the ability to react and adjust positively in times of adversity [52]. In some research, resilience has been operationalised as an inner strength and capacity, or one's ability to bounce back and recover after trauma, with a notion of accountability, but also through concepts of salutogenesis, coherence and social capital [52–55]. Individual and collective coping methods can be used to understand this dimension of resilience. In our study, resilience in this sense reflects individuals' ability to make use of family or community key persons as resources. For some informants, this led to conflicts with family members, social networks, and religious communities about how to understand and cope with the pandemic. Similar to other coping studies [56], our findings show that people's successful coping with stressors depends in part on social resources, which are in turn influenced by cultural values and norms in society.

But an overemphasis on individual and social levels of resilience tends to miss the awareness of power and agency dimensions. A subject is historically and culturally situated [57] and reacts to "a complex interplay of discourses, norms, power relations, institutions and practices" [55]. In light of this, when informants react with reluctance or suspicion, this is a result of historical and cultural narratives, be it genocide, hostile governments, or scapegoating. Or, when participating

in collective rituals during a pandemic, such as funerals or weddings, despite the risk, these events demonstrate a sense of agency. Informants demonstrate agency also when attempting to reproduce the information by authorities in their own mother tongue and through their own cultural TV-channels. The results of our study suggest that culturally and linguistically sensitive messages that consider people's social realities, not least those most vulnerable in society, can support their sense of purpose in changing their behaviour in a crisis situation, but also inform policy guidance in healthcare.

**Counteracting a syndemic through communication that reduces the impact of health inequalities.** The medical anthropological concept of syndemic refers to how diseases interact with and address the underlying social and environmental conditions [58]. The concept fits well with the synergetic interaction between the COVID-19 pandemic and social determinants observed in Sweden. A study of the direct and indirect effects of the COVID-19 pandemic on different social groups in Sweden shows that the relative risk of being affected by adverse events during the pandemic was strikingly similar across groups as it had been in the previous four years [59]. The social determinants of health that disproportionately affected vulnerable groups in society before the pandemic were also the most important factors at play during the COVID-19 pandemic. However, social determinants of health and their interaction with disease can be influenced. Our study suggests that well-functioning intercultural health communication can be an important tool in preventing a pandemic from automatically becoming a syndemic. Our results provide several examples of problems with intercultural communication during the COVID-19 pandemic and how these may have exacerbated the disproportionate impact of social determinants of health. Effective health communication that targets and actively involves populations with a history of health inequalities to ensure cultural and structural relevance is likely to help reduce the impact of health inequalities during a health crisis. The motivation, level of educational, family structure, structure of interactions between people and institutions, socio-economic conditions, cultural values and acculturation experiences of many of the participants are all examples of how multiple systems interact to reinforce and complicate the impact of a pandemic. Communicating crisis-related information in such a bio-socio-ecological context [60] therefore requires an awareness of more than the specific health-related issues.

## Reflexivity and trustworthiness

Language and culture were both a limitation and a strength of this study. Most of the interviews were conducted in Swedish, a second language for the participating informants, which occasionally caused language problems. Understanding was supported by careful repeated listening to ambiguous parts of the interviews. Reliability was strengthened by the researchers ÖC interviewing in Turoyo and BAA interviewing in Arabic and translating into Swedish. During the focus group interviews, the participants also helped each other translate and clarify the meaning of some terms, which not only contributed to our understanding as researchers, but also demonstrated their own agency and subjectivity in clarifying the issues. This helped to ensure that linguistic meanings were translated into both Swedish and English. Furthermore, the mixed linguistic and cultural background of the research team made it easier to reach informants for interviews and to analyse the data from different perspectives. The clear affiliation of the research project to the healthcare system was appreciated by the informants as it contributed to the level of trustworthiness and made participation meaningful. This is also linked to the complexity of the researchers' positioning, which reveals both strengths and limitations.

As mentioned earlier, two of our researchers had an insider position, which had the advantage of creating proximity and access to the different populations. At the same time, the outsider researchers were able to contribute with the necessary distance and curiosity also needed in analysing data without pre-assumptions. The insider position provided the opportunity to delve deeper into issues and to ask follow-up questions based on prior knowledge. The participation of outsider researchers was seen as a sign that the authorities were paying attention to the situation.

Another limitation is the higher age level of the participants. At the same time, this makes the vulnerability in communication more apparent than if we had a younger population with a better command of the Swedish language. The study population also included more women than men, which highlights the limitations of gender representativeness and also

the methodological challenge of reaching men for studies. The differences between the two areas should be taken into account, both in terms of historical and geographical backgrounds, religious differences, and levels of acculturation. However, in the initial stages of analysing we did not find any relevant differences that led us to analyse the two populations separately. In order to ensure rigour in the analysis process [61], the NVIVO 12 software program was used and in the first step of the analysis process, two researchers monitored the units of meaning, sub-themes, and themes until consensus was reached. This was followed by a third researcher and then the entire research team.

When conducting research on a sensitive topic among vulnerable populations, specific ethical considerations are paramount. Potential risks include stigmatization, breaches of integrity, and the triggering of traumatic memories. By revealing cultural differences during challenging societal happenings, there is a risk of unintentionally stigmatizing the very people we want to understand. Overemphasizing factors like ethnicity and religion can obscure other important variables, such as education, socio-economic status, age, and gender. To mitigate these risks, we contextualized our findings and sought to understand the respondents' stories from multiple perspectives. Our researchers' insider perspectives, combined with a rigorous scientific approach, align well with Benjamin Paul's philosophy of the basic significance of understanding the thinking of the community in order to drive positive change in public health, "If you wish to help a community improve its health, you must learn to think like the people of that community" [62]. To this we want to add the importance of local community involvement.

## Conclusion

For the non-native Swedish speaking participants in this study, living in underprivileged multicultural communities, understanding the COVID-19-related information from the authorities was a complicated process and the opportunity to receive information in their own language was important to facilitate understanding. What information to trust was a difficult question for many. Our findings point to the importance of language, community engagement, authorities working with local health facilities, community actors, trusted leaders and religious organisations, and the value of oral and culturally adapted information. We suggest that Antonovsky's concept of Sense of Coherence may be used for successful health crisis communication aimed at behavioural change. For the recipients, the communication needs to be comprehensible, manageable, and meaningful. The study findings also point to the importance of emphasising dialogue as a vital part of a health crisis communication framework, and the centrality of local community preparedness for future health crises. Further, the results suggest that culturally and linguistically sensitive messages that consider people's social realities, not least those most vulnerable in society, can support their sense of purpose in changing their behaviour in a crisis, but also inform policy guidance in healthcare.

### Implications and recommendations for future health crisis communication

For future pandemics or other health crises, structured preparation and planning of models for reaching all groups are urgently needed, instead of the ad hoc approach that was followed. The findings of this study mirror a growing body of research that highlights the importance of community engagement in emergency risk communication, in order to reach vulnerable populations with a history of health inequalities, social injustice, and limited access to health information. Further, that engagement begins before a crisis. [63–64]
*Suggestions on key aspects to be considered while communicating with linguistically diverse populations in future health crises*

- Continuous analysis of existing public health communication strategies, including language needs in populations.

- Continuously preparing for potential health-related crises, as diversity in the population and communication channels to reach subgroups keep evolving.

- Anchor communication in local and community specific trust-building strategies to reach vulnerable populations.

- Establish partnerships with local communities, individual or institutional, as part of national public health strategies and use these partnerships as soon as possible during health crises for effective dialogue and information dissemination.

- Be aware of power differences and admit the agency of local communities, by inviting them to prepare and involve themselves in crisis management and communication.

## Supporting information

**S1 Text. Interview questions.**
(DOCX)

## Acknowledgments

First and foremost, we want to express our gratitude to all participating residents who generously shared their experiences. We are also grateful to Fatuma Mohamed for her valuable assistance in reaching residents in the Järva area, Rabita Kerimo for valuable assistance in reaching participants in Södertälje, Mehrnaz Aram for information about the work of Region Stockholm. Furthermore, we want to thank Maria Pia Hergens at the Regional Dept. Of Communicable Disease Control and Prevention for information and our project reference group for their valuable input—thank you Maria Albin, Jesper Ekberg, Henrik Malm Lindberg, and Tanja Viklund.

## Author contributions

**Conceptualization:** Sofie Bäärnhielm, Soorej Jose Puthoopparambil, Mattias Strand, Önver Cetrez.

**Data curation:** Sofie Bäärnhielm, Baidar Al-Ammari, Soorej Jose Puthoopparambil, Mattias Strand, Önver Cetrez.

**Formal analysis:** Sofie Bäärnhielm, Baidar Al-Ammari, Soorej Jose Puthoopparambil, Mattias Strand, Önver Cetrez.

**Funding acquisition:** Sofie Bäärnhielm, Soorej Jose Puthoopparambil, Mattias Strand, Önver Cetrez.

**Investigation:** Sofie Bäärnhielm, Baidar Al-Ammari, Soorej Jose Puthoopparambil, Mattias Strand, Önver Cetrez.

**Methodology:** Sofie Bäärnhielm, Baidar Al-Ammari, Soorej Jose Puthoopparambil, Mattias Strand.

**Project administration:** Sofie Bäärnhielm.

**Resources:** Sofie Bäärnhielm.

**Software:** Sofie Bäärnhielm.

**Supervision:** Sofie Bäärnhielm, Önver Cetrez.

**Validation:** Sofie Bäärnhielm, Baidar Al-Ammari, Soorej Jose Puthoopparambil, Mattias Strand, Önver Cetrez.

**Visualization:** Sofie Bäärnhielm, Baidar Al-Ammari, Soorej Jose Puthoopparambil, Mattias Strand, Önver Cetrez.

**Writing – original draft:** Sofie Bäärnhielm, Baidar Al-Ammari, Soorej Jose Puthoopparambil, Mattias Strand, Önver Cetrez.

**Writing – review & editing:** Sofie Bäärnhielm, Baidar Al-Ammari, Soorej Jose Puthoopparambil, Mattias Strand, Önver Cetrez.

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
