## [Decision Letter · Decision Letter 0]

25 Aug 2025

PGPH-D-25-00670

How to make sense of information about COVID-19 and the vaccine from the authorities? A qualitative study of migrants' experiences in two Swedish communities

Dear Dr. Bäärnhielm,

Thank you for submitting your manuscript to PLOS Global Public Health. After careful consideration, we feel that it has merit but does not fully meet PLOS Global Public Health’s publication criteria as it currently stands. Therefore, we invite you to submit a revised version of the manuscript that addresses the points raised during the review process.

We look forward to receiving your revised manuscript.

Kind regards,

Myung-Bae Park

Guest Editor

Journal Requirements:

1. Please provide additional details regarding participant consent. In the ethics statement in the Methods and online submission information, please ensure that you have specified (1) whether consent was informed and (2) what type you obtained (for instance, written or verbal, and if verbal, how it was documented and witnessed).

2. Please provide additional information regarding the considerations made for the migrants included in this study. For instance, please discuss whether participants were able to opt out of the study and whether individuals who did not participate receive the same treatment offered to participants.

3. We have amended your Competing Interest statement to comply with journal style. We kindly ask that you double check the statement and let us know if anything is incorrect.

4. In the online submission form, you indicated that “Availability of data and materials

The datasets used during the current study are available from the corresponding author on reasonable request.”.

a. In a public repository,

b. Within the manuscript itself, or

c. Uploaded as supplementary information.

Additional Editor Comments (if provided):

Your manuscript has now been reviewed. We have received two reviewer reports. One reviewer recommends acceptance, while the other recommends minor revision. Based on these recommendations, we have reached the decision that your manuscript can be accepted for publication after minor revisions.

Reviewers' comments:

Reviewer's Responses to Questions

**Comments to the Author**

1. Does this manuscript meet PLOS Global Public Health’s publication criteria?

Reviewer #1: Yes

Reviewer #2: Yes

2. Has the statistical analysis been performed appropriately and rigorously?

Reviewer #1: N/A

Reviewer #2: N/A

3. Have the authors made all data underlying the findings in their manuscript fully available (please refer to the Data Availability Statement at the start of the manuscript PDF file)?

Reviewer #1: No

Reviewer #2: Yes

4. Is the manuscript presented in an intelligible fashion and written in standard English?

Reviewer #1: Yes

Reviewer #2: Yes

Reviewer #1: This paper examines how immigrant communities understood the Swedish government's COVID communications. The authors highlight several challenges, particularly language barriers, mistrust of the government, religion etc as some of the issues that migrants struggled with. The authors interviewed 76 individuals, of varying ages. Overall, the paper is clearly written although it is extremely long and could benefit from being shorter so that the key points being made are easier to grasp. The discussion, in particular, is much too long even for a qualitative paper. In terms of methodological rigor, the study is well designed and it is clear that the authors collected a lot of information. It also helps that some of the authors could speak some of the languages represented by the migrant communities they interviews.

Reviewer #2: A study addressing the experiences of migrants and linguistic minority groups during the global health crisis of the COVID-19 pandemic provides an important message to the international academic community. By interviewing 76 migrants with diverse ages, genders, and language backgrounds, the study successfully captures multiple perspectives.

However, the manuscript would become a more meaningful contribution if the following points are addressed:

Sample composition: Detailed information on participants’ socioeconomic backgrounds (e.g., education, occupation, length of residence) is limited. Clarifying how these background variables influence the understanding of COVID-19 information would strengthen the credibility of the analysis. In the same vein, indicating the interviewees’ age, gender, and language background alongside quotations would help readers better understand the context.

Policy implications: While the conclusion highlights the importance of community preparedness, it would be valuable to specify concrete implementation strategies (e.g., multilingual crisis communication manuals, training programs for health leaders, rapid translation systems) to enhance the study’s policy relevance.

**Do you want your identity to be public for this peer review?** For information about this choice, including consent withdrawal, please see our Privacy Policy

Reviewer #1: No

Reviewer #2: No

---

## [Editor Report · Decision Letter 1]

26 Nov 2025

How to make sense of information about COVID-19 and the vaccine from the authorities? A qualitative study of migrants' experiences in two Swedish communities

PGPH-D-25-00670R1

Dear Doctor Bäärnhielm,

We are pleased to inform you that your manuscript 'How to make sense of information about COVID-19 and the vaccine from the authorities? A qualitative study of migrants' experiences in two Swedish communities' has been provisionally accepted for publication in PLOS Global Public Health.

Best regards,

Myung-Bae Park

Guest Editor